# Humanized β2 Integrin-Expressing Hoxb8 Cells Serve as Model to Study Integrin Activation

**DOI:** 10.3390/cells11091532

**Published:** 2022-05-03

**Authors:** Thomas Bromberger, Sarah Klapproth, Markus Sperandio, Markus Moser

**Affiliations:** 1Institute of Experimental Hematology, School of Medicine, Technische Universität München, D-81675 Munich, Germany; thomas.bromberger@tum.de (T.B.); sarah.klapproth@tum.de (S.K.); 2Institute of Cardiovascular Physiology and Pathophysiology, Walter Brendel Center of Experimental Medicine, Ludwig-Maximilians-Universität München, D-82152 Planegg-Martinsried, Germany; markus.sperandio@lrz.uni-muenchen.de

**Keywords:** β2 integrin activation, KIM127, mAb24, Hoxb8 cells, talin1, kindlin3

## Abstract

The use of cell-based reporter systems has provided valuable insights into the molecular mechanisms of integrin activation. However, current models have significant drawbacks because their artificially expressed integrins cannot be regulated by either physiological stimuli or endogenous signaling pathways. Here, we report the generation of a Hoxb8 cell line expressing human β2 integrin that functionally replaced the deleted mouse ortholog. Hoxb8 cells are murine hematopoietic progenitor cells that can be efficiently differentiated into neutrophils and macrophages resembling their primary counterparts. Importantly, these cells can be stimulated by physiological stimuli triggering classical integrin inside-out signaling pathways, ultimately leading to β2 integrin conformational changes that can be recorded by the conformation-specific antibodies KIM127 and mAb24. Moreover, these cells can be efficiently manipulated via the CRISPR/Cas9 technique or retroviral vector systems. Deletion of the key integrin regulators talin1 and kindlin3 or expression of β2 integrins with mutations in their binding sites abolished both integrin extension and full activation regardless of whether only one or both activators no longer bind to the integrin. Moreover, humanized β2 integrin Hoxb8 cells represent a valuable new model for rapidly testing the role of putative integrin regulators in controlling β2 integrin activity in a physiological context.

## 1. Introduction

Regulated cell adhesion is a prerequisite for many physiological processes, such as the selective emigration of leukocytes from the vascular system, their migration within tissues, or the aggregation of platelets. The integrin family of adhesion receptors plays a central role in these processes, because they can change their adhesive properties by altering their conformational state in response to extracellular cues [1,2,3]. This characteristic feature, also called integrin activation, is particularly important and best studied in blood cells, especially for platelet αIIbβ3 and leukocyte β2 integrins, because their affinity for ligand increases dramatically upon activation [4,5,6,7]. Integrins on blood cells can adopt different conformations: a bent, inactive conformation with low ligand binding affinity that can transition into an extended integrin structure with intermediate affinity, and a fully active high-affinity conformation with an “open” ligand binding headpiece [1,7,8]. Recently, a high-affinity bent β2 integrin conformation binding ligands in cis has also been reported [9]. Ligand binding not only depends on changes in integrin affinity, but is also largely determined by subsequent events, such as clustering of integrins, adhesion structure assembly, and linkage to the actin cytoskeleton, which altogether significantly contribute to the stability or lifetime of the integrin–ligand interaction [10]. Therefore, the phenotype of reduced cell adhesion can be attributed to defects at different levels, and a clear molecular separation between integrin activation from later events is often difficult, and would require more sophisticated electron microscopy or crystallization approaches to monitor the change in integrin conformation. Since genetic manipulations of primary leukocytes and platelets are less amenable, cell-based reporter systems such as Chinese hamster ovary (CHO) cells stably expressing wild-type or mutant human platelet αIIbβ3 integrin became a frequently used tool to study integrin activation [11,12,13,14,15] since the transition towards an active αIIbβ3 integrin conformation can be detected by flow cytometry with the conformation-specific PAC-1 antibody [16]. Another advantage of these cells is the convenience of manipulating them by protein overexpression or knockdown experiments.

A multitude of in vitro and in vivo studies have shown in the past 20 years that all intracellular signaling pathways that induce integrin activation culminate in the binding of talin and kindlin proteins to the integrin cytoplasmic domain [3,17,18,19,20]. The CHO reporter cell system was indeed the first to demonstrate that overexpressed talin head strongly activates αIIbβ3 integrin [21,22] and that kindlins significantly potentiate talin head-induced integrin activation [17,23]. However, CHO cells do not respond to agonists as platelets or leukocytes do, and thus, further genetic manipulations have been made to obtain insights into the signaling cascades that regulate integrin activation [13,15]. Comparable to this, K562 cells, a human erythroleukemic cell line, were frequently used in the past to study β2 integrin activity [24,25,26,27], also because conformation-specific antibodies against the human αLβ2 integrin in its extended intermediate and fully active high-affinity conformation (Kim127 and mAb24) exist [28,29,30]. However, similarly to CHO cells, which naturally do not express αIIbβ3 integrin, K562 cells do not express β2 integrins and so do not properly respond to activation triggers [27,31]. In conclusion, the currently available cell systems to study integrin activation by conformation-specific antibodies have their limitations.

More recently, the Hoxb8 cell system has been established, which represents a murine hematopoietic progenitor cell line that can be efficiently differentiated into various myeloid and even lymphoid cell types in vitro and in vivo [32]. Importantly, the differentiated cell types functionally resemble their primary counterparts. Moreover, these cells can be easily generated from the bone marrow of any existing mouse strain and can be effectively manipulated by viral transduction and CRISPR-mediated genome editing [33,34].

In the current study, we humanized murine Hoxb8 cells from β2 integrin-deficient mice by stably expressing the human β2 integrin ortholog, which functionally fully replaced the murine counterpart in PMN-like cells and macrophages. We show that humanized β2 PMN-like cells can be activated by various agonists such as TNFα, CXCL1, fMLP, or PMA that trigger integrin inside-out signaling pathways, leading to integrin activation, which can be detected by KIM127 and mAb24 binding assays. Moreover, these cells can be efficiently manipulated via the CRISPR/Cas system or by introducing specific β2 integrin mutants via retroviral vectors. Using this system, we can show that kindlin3 is required for KIM127 and mAb24 binding, similar to talin1. We can further demonstrate that, in addition to talin1 and kindlin3, among the proteins tested, only the key talin regulators Rap1b and RIAM are critical for αLβ2 integrin activation, while other integrin regulatory proteins such as filamin, ILK, RapL, and Sharpin are not required. Thus, this cell system allows rapid testing of mutant β2 integrins or studying the role of potential integrin regulators in inducing conformational changes of β2 integrins under physiological conditions.

## 2. Materials and Methods

### 2.1. Plasmids

The retroviral vectors pMSCVneo-ER-Hoxb8 and pMIGR and the packaging plasmid pCl-Eco were gifts from Marc Schmidt-Supprian. Mouse or human integrin β2 cDNA were purchased from Addgene (Watertown, MA, USA) and subcloned into pMIGR via XhoI and SalI restriction sites. Mutations were introduced into the β2 integrin cytoplasmic domain using the QuikChange II site-directed mutagenesis kit (Agilent, Santa Klara, CA, USA).

### 2.2. Cell Line Generation and Cell Culture

Hoxb8 cells were generated from bone marrow of integrin β2^−/−^ and wild-type control mice as described previously [32]. Briefly, bone marrow was harvested and subjected to Ficoll-Paque (GE-Healthcare, Chicago, IL, USA) gradient centrifugation. The supernatant, containing lymphocytes and HPCs (hematopoietic progenitor cells), was collected and cultured for three days in stem cell medium (RPMI supplemented with 15% FCS, 1% penicillin/streptomycin, 500 µM L-glutamine (all from Thermo Fisher Scientific, Darmstadt, Germany), 10 ng/mL rmIL-3 (recombinant mouse Interleukin-3), 20 ng/mL rmIL-6 (both from Peprotech, Hamburg, Germany) and 2% SCF-containing supernatant from CHO cells). Cells were infected with ER-Hoxb8 (estrogen receptor–Hoxb8) virus by spin inoculation and further cultured in progenitor outgrowth medium (POM; RPMI1640 supplemented with 10% FCS, L-glutamine, penicillin/streptomycin, 50 µM β-mercaptoethanol, 1 µM β-estradiol (Sigma-Aldrich, Deisenhofen, Germany) and 5% Flt3 ligand-containing supernatant from a Flt3 ligand-producing B16 melanoma cell line). After 4 weeks of culture, all outgrown cells were immortalized progenitors. Established Hoxb8 cell lines were cultured in POM at standard cell culture conditions (37 °C, 5% CO_2_).

For differentiation into neutrophil-like cells, 1 Mio Hoxb8 cells was washed twice with PBS/2% FCS to fully withdraw β-estradiol and cultured in 40 mL R10 medium (RPMI1640, 10% FCS, L-glutamine, penicillin/streptomycin, 50 µM β-mercaptoethanol) supplemented with 3% SCF-containing supernatant and 20 ng/mL G-CSF (Peprotech) in a 15 cm cell culture dish for 4 days. Macrophage differentiation was initiated by culturing 1 Mio washed Hoxb8 cells in R10 medium supplemented with 10% MCSF-containing supernatant from L929 cell and 5% Flt3 ligand-containing supernatant overnight in a 10 cm Petri dish. Cells were split into 5–10 Petri dishes and cultured in R10 medium containing 10% MCSF for another 5–6 days.

### 2.3. CRISPR/Cas9-Mediated Gene Targeting

The CRISPR/Cas9 technique was used to ablate various genes in Hoxb8 cells. Guide RNAs (gRNAs) were designed using the online platform CHOPCHOP [35] and purchased from Integrated DNA Technologies (Leuven, Belgium). Ribonucleoprotein (RNP) complexes were allowed to form in vitro by incubation of gRNA with Truecut Cas9 Protein v2 (Thermo Fisher Scientific) at room temperature and subsequently delivered into Hoxb8 cells by electroporation using the NEON transfection system (Thermo Fisher Scientific) according to the manufacturer’s instructions. Targeting efficiencies were analyzed by sequencing and TIDE analysis [36] and confirmed by Western blotting.

The following gRNA sequences were chosen to target the genes encoding the respective proteins: integrin β2: GCGCAATGTCACGAGGCTGC, talin1: GGATCCGCTCACGAATCATG, kindlin3: AGAGTCGATATAGTCCCCGG, Rap1: GACGAGCTTATATTCACGCA, Riam: GTGTAGTTAAACTCTTCTCG, ILK: ACGCGGTGGCGGTGCGCTTG, paxillin: TGAACTTGACCGGCTGTTAC, filaminA: CTTACTAGGTGAAGGCACGT, RapL: CGGGCACCGTCACTGGCCGT, sharpin: GAAATGTCGCCGCCCGCCGG.

### 2.4. Virus Generation and Viral Infections

Viral particles for Hoxb8 cell generation and integrin β2 expression were generated by transfecting HEK293 cells with pMSCVneo-ER-Hoxb8 or a pMIGR containing mouse or human integrin β2 cDNA (purchased from Addgene, Watertown, MA, USA), respectively, together with the packaging plasmid pCl-Eco using Lipofectamine 2000 (Thermo Fisher Scientific). Virus-containing supernatants were harvested 2 days after transfection.

Cells were infected by spin inoculation. Therefore, 1–3 × 10^5^ cells in 500 µL POM were mixed with 500 µL virus-containing supernatant in the presence of Lipofectamine (1:1000) and centrifuged at 1000× *g* for 60 min.

### 2.5. Static Adhesion and Spreading Assays

Static adhesion and spreading assays with neutrophil- and macrophage-like cells were performed as previously described [37,38]. For adhesion assays, non-cell culture-treated 96-well plates were coated with 5 µg/mL fibronectin (Merck Millipore, Darmstadt, Germany) or 4 µg/mL rmICAM1 (R&D systems) in coating buffer (20 mM Tris-HCl pH 9.0, 150 mM NaCl, 2 mM MgCl_2_) overnight. Plates were blocked with 3% BSA. Neutrophil-like cells (PMN-LCs) were either left unstimulated or activated with 20 ng/mL TNFα (R&D systems, Minneapolis, MN, USA) or 0.1 µg/mL PMA (Merck Millipore) immediately before seeding. Then, 5 × 10^4^ cells resuspended in adhesion medium (RPMI1640 supplemented with 0.1% FCS) were plated per well. Cells were washed and fixed with 4% PFA after 30 min. Adherent neutrophils were stained with DAPI, imaged using an EVOS M7000 life cell imaging system (Thermo Fisher Scientific), and counted using ImageJ software (National Institute of Health, Bethesda, MD, USA). Adherent macrophage-like cells were stained with crystal violet (5 mg/mL in 2% ethanol). After washing, dye retained by the cells was solved in 2% SDS and quantified by measuring absorption at 595 nm with a plate reader.

Spreading assays were performed in ICAM1 or fibronectin-coated non-cell culture treated 24-well plates; 10 cells within each of 3 different fields of view were measured per condition.

### 2.6. Flow Chamber Assays

Rolling and adhesion of PMN-LCs under flow conditions were assessed in ibidi slides VI0.1 (ibidi, Gräfelfing, Germany). Slides were coated with rmP-Selectin (His-tagged, R&D systems), mouse soluble ICAM1 (STEMCELL Technologies, Vancouver, Kanada), and rmKC (R&D systems) in coating buffer overnight. Hoxb8-derived PMN-LCs were resuspended in adhesion medium at a density of 7.5 × 10^5^ cells/mL and perfused through flow chambers using a PHD ULTRA pump (Harvard Apparatus, Holliston, MA, USA) generating a wall shear stress of 1 dyne/cm^2^. After 10 min of perfusion, 10 s time lapse movies were acquired from 5 different fields of view using an EVOS M7000 life cell imaging system. To assess integrin αLβ2 independent neutrophil rolling, cells were pre-incubated with rat anti-mouse CD11a (integrin αL)-blocking antibody (clone: M17/4, eBioscience). Velocities of rolling cells and the number of adherent cells were analyzed using ImageJ software. Up to 10 rolling cells were measured per field of view.

### 2.7. Adhesion Signaling

Adhesion signaling was in essence performed as described earlier [39]. Briefly, Hoxb8-derived macrophages were trypsinized and serum-starved in adhesion medium for 2 h in suspension. Subsequently, 8 Mio cells were pelleted and directly lysed in MPER (Mammalian protein extraction reagent) buffer (Thermo Fisher Scientific) supplemented with protease and phosphatase inhibitors. Another 8 Mio cells were lysed 20 min after plating onto an ICAM1-coated 10 cm dish. Protein lysates were subjected to Western blot analysis.

### 2.8. Flow Cytometry and Cell Sorting

Cells were stained for flow cytometry and sorting following standard procedures. Briefly, cells were incubated with Fc receptor-blocking antibody for 10 min and live/dead stain (Promofluor-840 maleimide; PromoCell, Heidelberg, Germany) and subsequently stained with fluorophore-conjugated antibodies for 30 min in FACS buffer (PBS supplemented with 2% FCS and 2 mM EDTA). Flow cytometry measurements were acquired using a Cytoflex LX flow cytometer (Beckman Coulter, Brea, CA, USA). A FACS Aria II Flow Cytometry Cell Sorter (BD Bioscience, Heidelberg, Germany) was used for cell sorting.

### 2.9. β2 Integrin Activation Assay

Mouse integrin β2-deficient and human integrin β2-expressing Hoxb8 cells were differentiated into PMN-LCs. After incubation with Fc receptor-blocking antibody and live/dead stain, cells were either left untreated or treated with 10 mM EDTA, 0.2 µg/mL TNFα, 10 µM fMLP, 1 µg/mL CXCL1, or 1 µg/mL PMA for 30 min at 37 °C in adhesion medium in the presence of BV421-conjugated active human integrin β2 conformation-specific antibody mAb24 or Alexa Fluor 647-conjugated extended human integrin β2 conformation-specific antibody KIM127. In one set of experiments, cells were preincubated with 10 µM Wortmannin, U73122, or Gö6983 (all from Sigma-Aldrich) for 10 min at 37 °C before addition of the antibody. Staining intensities were assessed using a Cytoflex LX flow cytometer. All data were normalized to total human integrin β2 surface levels.

KIM127 antibody was conjugated to Alexa Fluor 647 using an Alexa Fluor 647 antibody labeling kit (Thermo Fisher Scientific) following the manufacturer’s instructions.

### 2.10. Statistical Analysis

Data are shown as mean ± standard deviation (SD). Statistical significance was tested by one-way or two-way ANOVA followed by Tukey or Sidak’s multiple comparison test using Prism9 (GraphPad Software; San Diego, CA, USA). Differences between groups were assumed to be statistically significant if *p* < 0.05.

## 3. Results

To establish a Hoxb8 cell line expressing human instead of murine β2 integrin, we immortalized progenitor cells from the bone marrow of β2 integrin-deficient mice [40] by retroviral expression of an estrogen-regulated Hoxb8 transcription factor [32]. After establishment of this cell line, we transduced these cells again with retroviral vectors expressing either human or mouse β2 integrin cDNAs and finally sorted for β2 integrin-positive (human or mouse) Hoxb8-β2^−/−^ cells (Hoxb8-β2^−/−^/hβ2 and Hoxb8-β2^−/−^/mβ2) (Figure 1A). The presence of human β2 integrin on the cell surface already suggests its capability to form heterodimers with the appropriate murine α subunits. To confirm this, both cell lines as well as Hoxb8-β2^−/−^ and Hoxb8 cells that were generated from wild-type mice (Hoxb8-β2^+/+^) were differentiated into neutrophil-like cells (PMN-LCs) expressing Gr-1 and PSGL-1 (Figure 1B) and analyzed for their αL and αM integrin surface expression by flow cytometry. As expected, Hoxb8-β2^−/−^ PMN-LCs showed no αL and αM surface expression, whereas their surface levels were comparable between Hoxb8-β2^+/+^ and both Hoxb8-β2^−/−^/hβ2 and Hoxb8-β2^−/−^/mβ2 PMN-LCs, pointing towards similar and physiological human and mouse β2 integrin expression levels (Figure 1B). Of note, expression levels of human β2 on Hoxb8-β2^−/−^/hβ2 PMN-LCs were similar to that of primary human neutrophils (Figure 1C). Next, we tested human β2 integrin function in Hoxb8 cell-derived murine PMN-LCs. Static adhesion assays on ICAM1-coated surfaces upon activation with TNFα and PMA revealed similar adhesion of Hoxb8-β2^−/−^/hβ2-derived PMN-LCs compared to Hoxb8-β2^−/−^/mβ2 and Hoxb8-β2^+/+^ cell-derived PMN-LCs, whereas Hoxb8-β2^−/−^ cells failed to adhere (Figure 1D). We then analyzed αLβ2 integrin-mediated neutrophil slow rolling and adhesion on P-selectin-, ICAM1-, and CXCL1-coated flow chambers. In contrast to β2 integrin-deficient PMN-LCs that showed strongly increased rolling velocities and failed to adhere, both Hoxb8-β2^−/−^/hβ2 and Hoxb8-β2^−/−^/mβ2-derived PMN-LCs showed significantly slower rolling velocities similar to wild-type PMN-LCs (Figure 1F). Moreover, CXCL1-induced adhesion under flow was comparable between human and mouse β2 integrin-rescued PMN-LCs but slightly less than PMN-LCs differentiated from Hoxb8-β2^+/+^ cells (Figure 1E,G). In sum, these data indicate that expression of human β2 integrin can functionally replace murine β2 integrin in Hoxb8 cell-derived PMN-LCs.

Next, we analyzed cell adhesion, spreading, and β2 integrin-mediated signaling in Hoxb8 cell-derived macrophages that express either mouse or human β2 integrin. These cells and macrophages differentiated from Hoxb8-β2^+/+^ cells showed, like PMN-LCs, comparable integrin surface levels, whereas flow cytometry confirmed no αL and αM integrin surface expression on β2 integrin-deficient macrophages (Appendix A). As expected, macrophage adhesion to the β1 and β3 integrin ligand fibronectin was comparable between all cells, and both Hoxb8-β2^−/−^/hβ2- and Hoxb8-β2^−/−^/mβ2-derived macrophages showed a similar adhesion to ICAM1 as macrophages differentiated from Hoxb8-β2^+/+^ cells, whereas Hoxb8-β2^−/−^-derived macrophages failed to adhere to ICAM1 (Figure 2A). Furthermore, no difference in cell spreading on fibronectin and ICAM1 was measured, except for Hoxb8-β2^−/−^ macrophages that failed to spread on ICAM1 (Figure 2B,C).

An important prerequisite for similar regulatory and signaling mechanisms of human and mouse β2 integrins is that the intracellular domains are nearly identical, so that binding of mouse proteins to the cytoplasmic tail of the human β2 integrin is not affected, resulting in unimpaired signal transduction into the cell (Figure 2D). To test this experimentally, we analyzed β2 integrin-mediated adhesion signaling by plating starved macrophages on ICAM1-coated dishes for 20 min, then lysed the cells and analyzed general protein tyrosine phosphorylation (using the 4G10 antibody) as well as Pyk2, paxillin, and ERK1/2 phosphorylation by Western blotting. This experiment showed similar cell adhesion- and spreading-induced protein phosphorylation patterns in cells expressing either human or mouse β2 integrins, suggesting that β2 integrin-triggered signaling is not impaired in humanized β2 integrin Hoxb8 cell-derived macrophages (Figure 2E). This reconfirms that human β2 integrin forms functional heterodimers with murine α subunits that exhibit normal cell adhesion and signal transduction.

Having shown that human β2 integrin is fully functional in murine Hoxb8 cell-differentiated PMN-LCs and macrophages, we next wanted to test whether this cellular model system is appropriate to study β2 integrin activity regulation with the help of conformation-specific antibodies. In a first approach to address this, we differentiated Hoxb8-β2^−/−^/hβ2 and Hoxb8-β2^−/−^/mβ2 cells into PMN-LCs and stimulated them with TNFα, fMLP, and PMA before staining with KIM127 and mAb24 antibodies that recognize the extended and extended open β2 integrin conformation, respectively. As shown in Figure 3A,B, we detected robust binding of both KIM127 and mAb24 to the human β2 integrin by flow cytometry upon stimulation. In contrast, PMN-LCs expressing mouse β2 integrin showed no mAb24 binding and a weaker KIM127 binding response (Figure 3A,B). Importantly, the increased antibody binding was not due to stimulation-induced changes in αL, αM, and β2 integrin surface expression (Appendix A).

In a second approach, we wanted to investigate whether these cells are also suitable to provide insights into the upstream signaling cascades that regulate this process. Therefore, we stimulated Hoxb8-β2^−/−^/hβ2 PMN-LCs with the same agonists and simultaneously treated them with inhibitors of phosphatidylinositol-3-kinases (PI3K), phospholipase C (PLC), and protein kinase C (PKC), which are central signaling nodes that control β2 integrin activation. These experiments demonstrated that their inhibition significantly impaired β2 integrin activation in humanized PMN-LCs, as evidenced by decreased binding of KIM127 and mAb24 (Figure 3C,D). This finding indicates that these cells also serve as a useful model for studying integrin inside-out signaling pathways.

To further investigate the suitability of humanized β2 integrin Hoxb8 cells as a cell reporter system to test integrin activity, we measured mAb24 and KIM127 binding in the absence of the two key integrin regulators, talin1 and kindlin3. Therefore, we destroyed both genes in Hoxb8 cells by CRISPR/Cas9-mediated gene deletion. In a first experiment, we electroporated Hoxb8 cells with Cas9–RNP complexes targeting talin1 and kindlin3 genes to generate individual Hoxb8 cell clones that showed homozygous deletion of either talin1, kindlin3, or both genes, confirmed by sequencing and Western blot analyses. Wild-type and mutant Hoxb8 cell clones were then electroporated again with Cas9–RNP complexes targeting the Itgb2 gene and subsequently infected with a human ITGB2 cDNA carrying retrovirus before we selected mouse integrin β2-negative and human β2-positive cells by FACS (Figure 4A,B). Wild-type and mutant Hoxb8-β2^−/−^/hβ2 cell clones were then differentiated into PMN-LCs, stimulated with TNFα and PMA, and analyzed for KIM127 and mAb24 binding by flow cytometry. This experiment led to several interesting results. (i) To our surprise, kindlin3 deficiency not only impaired mAb24 binding, as previously shown, but also abolished KIM127 binding, which is in contrast to the previous view that kindlin3 is required for the induction of the high-affinity conformation but not for initial ectodomain extension [26,41]. Thus, under the conditions chosen here, upon stimulation with soluble agonists, both talin1 and kindlin3 are required for β2 integrin extension (Figure 4C,D). (ii) Loss of either talin1 or kindlin3 results in a similar defect in integrin activation. However, very strong activation with the non-physiological agonist PMA allowed some more integrins to become fully activated in the absence of kindlin3, as indicated by slightly increased mAb24 binding (Figure 4C). (iii) Combined deletion of both integrin activators does not further decrease integrin activity, indicating that both regulators are absolutely required for β2 integrin activation (Figure 4C,D). Accordingly, PMN-LCs lacking either kindlin3 or talin1 failed to adhere under flow on P-selectin-, ICAM1-, and CXCL1-coated surfaces, similar to PMN-LC lacking both kindlin3 and talin1 (Figure 5A). Moreover, under these conditions, their rolling velocities were significantly increased compared to wild-type PMN-LCs, although kindlin3-deficient PMN-LCs showed an intermediate phenotype and rolled slower than talin1-deficient PMN-LC (Figure 5B). Of note, the differences in rolling velocity between genotypes completely disappeared when cells were treated with LFA-1-blocking antibody prior to the experiment (Figure 5C). On the other hand, P-selectin-induced integrin-mediated slow rolling (in the absence of chemokine) was comparable between kindlin3-deficient and wild-type PMN-LCs, whereas talin1-deficient cells showed faster rolling (Figure 5D). These results suggest that P-selectin-induced integrin-mediated slow rolling is not affected by the absence of kindlin3, but additional chemokine-induced slower rolling that eventually leads to cell arrest is affected by the absence of kindlin3.

In a next step, we wanted to assess whether the Hoxb8 system can also be used to test the influence of specific mutations within the β2 integrin on its activation. To this end, we retrovirally expressed human β2 integrin constructs carrying mutations in the binding motifs of talin1 (F754A) and kindlin3 (TTT758/759/760AAA) within the integrin cytoplasmic tail in Hoxb8-β2^−/−^ cells (as shown in Figure 1A and Figure 6A). Cells with equal human β2 integrin surface levels were sorted, differentiated into PMN-LCs, and analyzed for mAb24 binding upon activation by TNFα, CXCL1, fMLP, and PMA. This experiment clearly showed that binding of talin1 and kindlin3 are required for β2 integrin activation (Figure 6B). Interestingly, when we normalized the mAb24 binding intensity to the β2 integrin surface level, we realized that β2 integrins with mutations in the kindlin3 binding site are expressed at much lower surface levels. Since the kindlin3 binding site is also recognized by the critical regulator of endosomal recycling SNX17, we believe that this phenomenon is due to reduced cell surface recycling and increased lysosomal degradation of the mutant integrin. Interestingly, expression of β2 integrins with a mutation in the kindlin3 binding membrane distal NPxF motif, which is also bound by SNX17, results in an even stronger reduction in surface levels (data not shown) [42,43]. This ancillary finding illustrates that human β2 integrin expressed in mouse cells is subject to normal turnover mechanisms.

Finally, we wanted to test whether the humanized β2 Hoxb8 cells can be used to rapidly screen proteins for their ability to regulate β2 integrin activation. To this end, we used a CRISPR/Cas9 gene deletion approach, where we first as a proof of concept electroporated Hoxb8-β2^−/−^/hβ2 cells with RNP complexes of Cas9 protein assembled with either a non-targeting control gRNA (NTC) or specific talin1 and kindlin3 gRNAs (Figure 7A). Bulk Hoxb8-β2^−/−^/hβ2 cells were then directly differentiated into PMN-LCs and high targeting efficiency was determined by sequencing of the targeted gene locus and Western blot analysis (Figure 7B,C). This fast screening approach also identified the crucial role of talin1 for integrin ectodomain extension and induction of the high-affinity conformation indicated by strongly reduced KIM127 and mAb24 binding compared to NTC-treated control cells. Again, kindlin3 targeting not only impaired mAb24 binding, but also strongly reduced KIM127 binding similar to talin1-targeted cells (Figure 7D,E). Using this approach, we then analyzed a set of proteins that have previously been proposed to regulate leukocyte integrin activity. These were ILK [44], FilaminA [45,46], Sharpin [47], RapL [48], Rap1b [49,50,51], and RIAM [38,52]. After RNP complex electroporation, high efficiency of gene deletion was confirmed by Western blotting (Figure 7F,G). The targeted Hoxb8-β2^−/−^/hβ2 cells were then differentiated into PMN-LCs and stimulated with different agonists before we measured mAb24 binding by flow cytometry. This experiment showed that only CRISPR-mediated deletion of the direct talin1 interactors Rap1b and RIAM significantly prevented the induction of β2 integrin conformational changes upon stimulation, whereas targeted deletion of ILK, RapL, Sharpin, and FilaminA showed no effect (Figure 7H).

## 4. Discussion

The main aim of the present study was to establish a cell system that allows rapid and simple analysis of β2 integrin conformational changes, defined as integrin activation, under physiological conditions. To this end, we used Hoxb8 cells, which are conditionally immortalized progenitor cells of the hematopoietic system. The expression of the Hoxb8 homeobox gene linked to a mutant estrogen receptor inhibits myeloid differentiation in the presence of estradiol and arrests these cells in a phenotype of multipotent progenitors. These cells can be expanded indefinitely and, upon estradiol withdrawal, differentiate into neutrophil-like cells, macrophages, or dendritic cells dependent on the growth conditions and cytokines they are exposed to [32]. Most importantly, these in vitro differentiated cells functionally mimic their primary counterparts, also demonstrated by their ability to engraft into live mice and be studied in in vivo settings [53]. Thus, the in vitro-generated Hoxb8 cell-derived myeloid cells behave similarly to primary cells, which is, in our opinion, the major benefit compared to other currently used cell systems to study integrin activation (CHO-A5, K562, HL-60 cells). Another important advantage of the Hoxb8 cell system is that the cells can be generated from bone marrow or fetal liver cells of virtually any existing mouse strain. This and the fact that Hoxb8 cells can be genetically manipulated very efficiently in their progenitor stage show also their advantage over approaches to study human primary cells [33,34].

Since conformation-specific antibodies only exist for human β2 integrin, we expressed human β2 integrin in Hoxb8 cells that were initially generated from β2 integrin-deficient mice. A comprehensive cell biological characterization revealed that β2 integrin replacement did not affect its function in neutrophil-like cells and macrophages. Comparable rolling and adhesive properties between PMNs expressing either murine or human β2 integrin indicated that a hybrid mouse/human αLβ2 integrin can adopt both the extended closed and extended open conformations. The nearly identical cytoplasmic domains of mouse and human β2 integrins suggested a normal regulation of human β2 integrin by murine cytoplasmic proteins, as evidenced by normal induction of cell adhesion and signaling. Most importantly, humanized β2 Hoxb8 cell-derived PMN-LCs responded to natural stimuli such as chemokines (CXCL1), cytokines (TNFα), or N-formyl oligopeptides (fMLP), which trigger the well-known integrin inside-out signaling cascade, leading to conformational changes of the β2 integrin that can be detected by the two conformation-specific antibodies Kim127 [30] and mAb24 [28].

Using the new humanized β2 Hoxb8 cells, we performed three experimental approaches. First, β2 integrin activation was analyzed in single cell clones generated by CRISPR/Cas9-mediated deletion of talin1, kindlin3, and both genes. Second, we expressed human β2 integrin mutants in β2 integrin-deficient Hoxb8 cells and analyzed β2 integrin activation upon stimulation. Finally, we investigated the role of a series of integrin-associated proteins on β2 integrin activation in a rapid screening approach. While the first approach is rather time consuming and costly and involves the risk of additional unrecognized genomic alterations within the selected clones, the latter application provides fast results but depends on high knockout efficiency and thus may require testing of multiple guide sequences. On the other hand, as mentioned above, Hoxb8 cells can easily be generated from already existing mouse strains. While these cell lines may show some variability between each generation process, clonal artifacts and variability due to unclear genetic status can be excluded here. Of course, viral expression of mutant proteins, as demonstrated by the expression of mutant β2 integrins, is an additional option. However, we would like to note that simple overexpression of, e.g., talin head, as typically performed in CHO-A5 cells, is not possible; first, the transfer of double-stranded DNA via lipofection or electroporation is not possible in the Hoxb8 cell system, and second, our retroviral expression systems were not able to reach the non-physiologically high talin head levels required for induction of integrin activity (data not shown).

With the humanized β2 integrin Hoxb8 cell system, we primarily investigated talin1- and kindlin3-mediated β2 integrin activation and demonstrated the simplicity of its manipulation. All three employed experimental approaches demonstrated that β2 integrin extension, indicated by KIM127 binding, is not possible under physiological stimulation when either talin1 or kindlin3 is absent or when binding of already one protein to the integrin tail is abolished. This finding is underpinned by the observation that talin1/kindlin3 double-deficient cells showed no further reduction in integrin activation. However, neutrophil rolling on P-selectin/ICAM1 is not affected by the absence of kindlin3, but is affected by the loss of talin1. This argues for a sequential regulation of β2 integrin during the rolling and adhesion process of leukocytes, in which the talin1-dependent, selectin-induced slow rolling velocity of leukocytes is further reduced by an additional chemokine-triggered, kindlin3-dependent signal that eventually leads to cell arrest [26]. Since kindlin3-deficient neutrophils do not show KIM127 binding but do show slow rolling, this suggests that either ectodomain extension is not required for slow rolling of leukocytes or that KIM127 binding to cells stimulated in solution allows no prediction of their rolling ability. Rather, our studies suggest that initial induction of ectodomain extension already requires talin1 and kindlin3 cooperativity, and also strongly opposes previous in vitro data that talin binding to the integrin tail is sufficient to induce integrin extension [54]. Consistent with this, impaired ectodomain extension in kindlin3-deficient neutrophils has also been recently reported [55]. How talin1 and kindlin3 cooperate in inducing ectodomain extension is not yet clear; however, a recent study proposed that transient binding of kindlin3 to the membrane proximal integrin tail region primes talin1-mediated unclasping of the integrin tails [56].

The humanized β2 integrin Hoxb8 system also allowed us to rapidly test the relevance of a number of proteins thought to have important roles in leukocyte integrin activation. These were the integrin tail-binding proteins FilaminA and Sharpin, which are known to bind to the β and αL subunit, respectively, and thereby inhibit integrin activation by preventing the binding of talin and kindlin to the integrin tail [45,46,47]. We also disrupted the Rap1 effector RapL, which has been shown to activate αLβ2 integrins in lymphocytes upon chemokine stimulation [48], and the well-known focal adhesion protein ILK, which has recently been proposed as a crucial β2 integrin regulator in neutrophils [44]. We also included the major neutrophil Rap1 protein, Rap1b, and the Rap1 effector RIAM in our studies. Indeed, only genetic deletion of the latter two direct talin interactors, Rap1b and RIAM, resulted in a significant decrease in mAb24 binding, likely due to reduced recruitment of cytoplasmic inactive talin to the plasma membrane in the course of integrin inside-out signaling [33,37,38,57]. Although high knockout efficiencies of 70% to 90% were also obtained for the other genes, we cannot completely exclude the possibility that the absence of a detectable phenotype is due to insufficient gene deletion. Nevertheless, our data strongly suggest that these proteins are not directly involved in regulating integrin conformational changes, but rather contribute to the assembly of the adhesion structure, its organization, cytoskeletal linkage, or its turnover.

## Figures and Tables

**Figure 1 cells-11-01532-f001:**
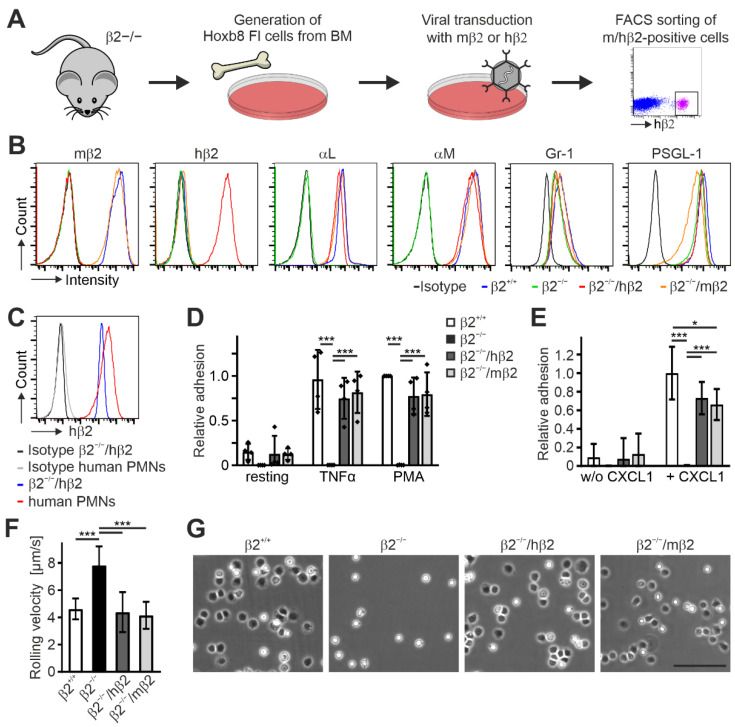
Integrin β2 can be substituted by its human homolog in mouse Hoxb8 FL cells. (**A**) Scheme of workflow. Hoxb8 FL cells were generated from bone marrow of integrin β2-deficient mice and retrovirally transduced with mouse or human integrin β2. Human/mouse integrin β2-positive cells were FACS-sorted. (**B**) Surface expression of different integrin subunits and cell markers on neutrophil-like cells differentiated from control (β2^+/+^), integrin β2 ko (β2^−/−^), and human or mouse integrin β2-rescued integrin β2^−/−^ (β2^−/−^/hβ2 and β2^−/−^/mβ2) Hoxb8 FL cells assessed by FACS analysis. (**C**) Integrin β2 surface expression levels on neutrophil-like cells differentiated from integrin β2^−/−^ Hoxb8 FL cells retrovirally transduced with human integrin β2 (β2^−/−^/hβ2) compared to PMNs isolated from human blood. (**D**) Static adhesion of untreated, TNFα-treated, or PMA-treated Hoxb8 FL-derived β2^+/+^, β2^−/−^, β2^−/−^/hβ2, and β2^−/−^/mβ2 neutrophils on ICAM1. N = 4. Individual data points of the 4 independent experiments are shown. (**E**,**F**) Adhesion (**E**) and rolling velocities (**F**) of neutrophil-like cells differentiated from β2^+/+^, β2^−/−^, β2^−/−^/hβ2, and β2^−/−^/mβ2 Hoxb8 FL cells in flow chambers coated with ICAM1 and P-selectin with or without CXCL1 under constant shear rate of 1 dyn/cm^2^. N = 10/11 chambers with CXCL1 (rolling and adhesion); N = 3/4 without CXCL1 (adhesion). (**G**) Neutrophil-like β2^+/+^, β2^−/−^, β2^−/−^/hβ2, and β2^−/−^/mβ2 cells plated on a P-selectin-, ICAM1-, and CXCL1-coated surface for 10 min. Scale bar: 100 µm. All values are given as mean ± SD. * *p* < 0.05, *** *p* < 0.001.

**Figure 2 cells-11-01532-f002:**
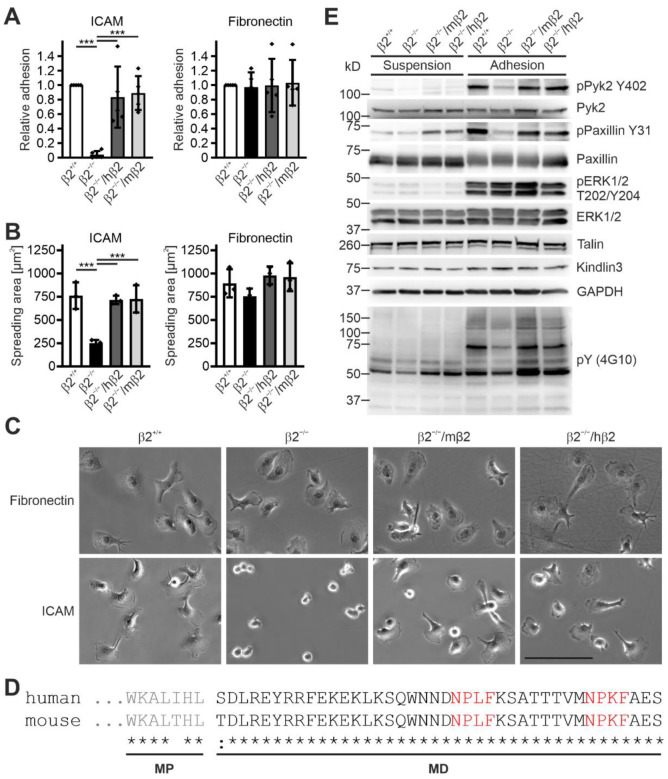
Expression of human integrin β2 rescues spreading and adhesion defects in mouse integrin β2 knockout macrophages. (**A**) Adhesion of integrin β2^−/−^ and human or mouse integrin β2 expressing integrin β2^−/−^ macrophages to ICAM1 and fibronectin in relation to control macrophages. N = 5. Individual data points represent the 5 independent experiments. (**B**) Spreading of control, integrin β2^−/−^, and human or mouse integrin β2 expressing integrin β2^−/−^ macrophages to ICAM1 and fibronectin assessed 2 h after plating. N = 3 independent experiments, shown as individual data points. (**C**) Hoxb8-derived β2^+/+^, β2^−/−^, β2^−/−^/hβ2, and β2^−/−^/mβ2 macrophages plated on an ICAM1- or fibronectin-coated surface for 2 h. Scale bar: 100 µm. (**D**) Alignment of the amino acid sequence of the mouse and human β2 integrin cytoplasmic tails. Talin-binding NPLF and kindlin-binding NPKF motives are highlighted in red. MP, membrane proximal; MD, membrane distal. (**E**) Control, integrin β2^−/−^, and human or mouse integrin β2 expressing integrin β2^−/−^ macrophages were either kept in suspension or plated on ICAM1 for 20 min to assess integrin-mediated signaling. Western blot analyses for Y402-phosphorylated and total Pyk2; Y31-phosphorylated and total paxillin; T202/Y204-phosphorylated and total ERK1/2; talin; kindlin3; and general tyrosine phosphorylation. GAPDH served as loading control. All values are given as mean ± SD. *** *p* < 0.001.

**Figure 3 cells-11-01532-f003:**
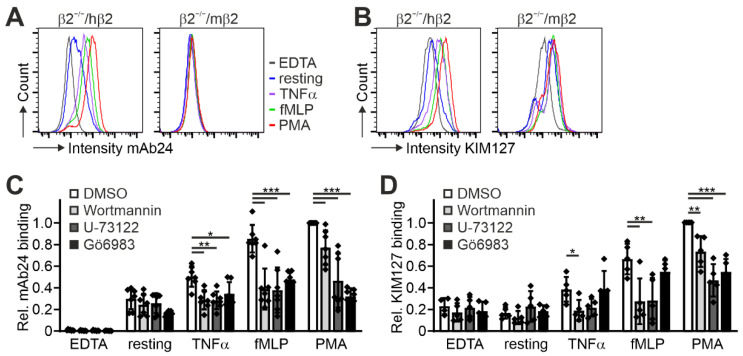
Expression of human integrin β2 in mouse integrin β2^−/−^ Hoxb8 cells allows assessment of integrin activity using conformation-specific anti-human integrin β2 antibodies. (**A**,**B**) FACS analyses of integrin β2 activation of Hoxb8-derived neutrophils expressing human or mouse integrin β2 by measuring staining intensities of conformation-specific antibodies mAb24 (**A**) and KIM127 (**B**) either untreated, EDTA-treated, or in response to TNFα, fMLP, or PMA. (**C**,**D**) Relative mAb24 (**C**) and KIM127 (**D**) binding to resting or EDTA-, TNFα-, fMLP-, or PMA-stimulated Hoxb8-derived neutrophil-like cells upon treatment with DMSO, the phosphatidylinositol-3-kinase (PI3K) inhibitor Wortmannin, the phospholipase C (PLC) inhibitor U-73122, or the protein kinase C (PKC) inhibitor Gö6983. N = 7 (mAb24) or 5 (KIM127) independent experiments, shown as individual data points. All values are given as mean ± SD. * *p* < 0.05, ** *p* < 0.01, *** *p* < 0.001.

**Figure 4 cells-11-01532-f004:**
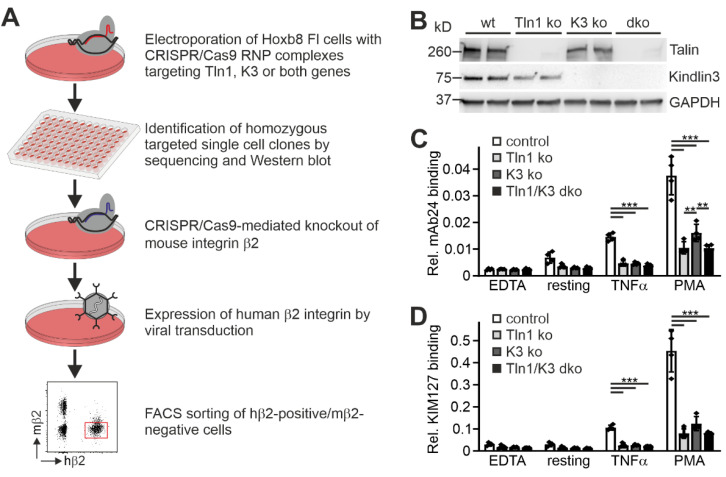
Both talin1 and kindlin3 are required for β2 integrin activation. (**A**) Scheme of workflow. Talin1, kindlin3, and talin1/kindlin3 double-deficient Hoxb8 FL cells were generated via the CRISPR/Cas9 system. Single cell clones were screened for talin1 and kindlin3 expression. Four clones per genotype were subjected to CRISPR/Cas9-mediated integrin β2 ablation and retrovirally transduced to express human integrin β2. Mouse integrin β2-negative and human integrin β2-positive cells were FACS-sorted. (**B**) Western blot analysis of neutrophil-like cells differentiated from different human integrin β2 expressing Hoxb8 single cell clones, in which talin1 and/or kindlin3 were ablated with the CRISPR/Cas9 system. GAPDH served as loading control. (**C**,**D**) Relative mAb24 (**C**) and KIM127 (**D**) binding to neutrophil-like cells derived from Hoxb8 single cell clones, expressing human integrin β2 and lacking talin1 and/or kindlin3 expression. N = 4 clones. Result of each clone is plotted as individual data point. All values are given as mean ± SD. ** *p* < 0.01, *** *p* < 0.001.

**Figure 5 cells-11-01532-f005:**
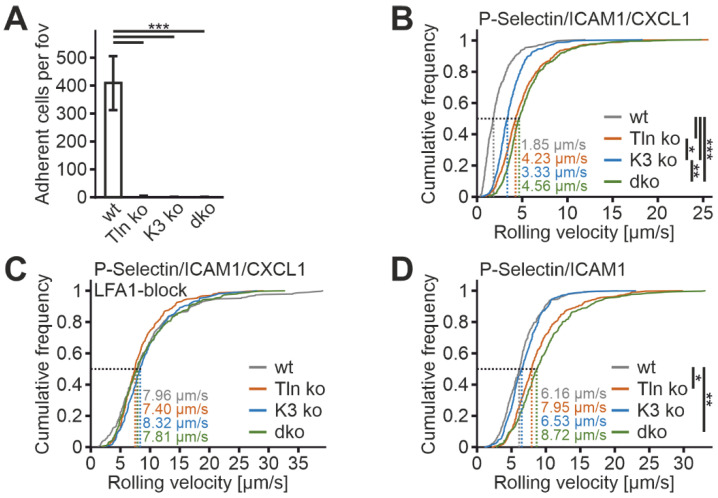
Kindlin3 is dispensable for P-selectin-induced integrin αLβ2-mediated slow rolling but required for chemokine-induced slower rolling. (**A**–**C**) Adhesion (**A**), rolling velocities (**B**), and rolling velocities after LFA-1 blocking (**C**) of Hoxb8 cell-derived PMN-LCs in flow chambers coated with ICAM1, P-selectin, and CXCL1 under constant shear rate of 1 dyn/cm^2^. N = 12–16 (**A**,**B**) and 6–9 (**C**) flow chambers. (**D**) Rolling velocities of PMN-LCs on ICAM1- and P-selectin-coated surfaces (without CXCL1) under constant shear rate of 1 dyn/cm^2^. N = 8–9 flow chambers. Cells were differentiated from different single cell clones, in which talin1 and/or kindlin3 were ablated with the CRISPR/Cas9 system. Rolling velocities are shown as cumulative distribution of the velocities of approximately 500 (**B**), 300 (**C**), and 400 (**D**) cells. * *p* < 0.05, ** *p* < 0.01, *** *p* < 0.001.

**Figure 6 cells-11-01532-f006:**
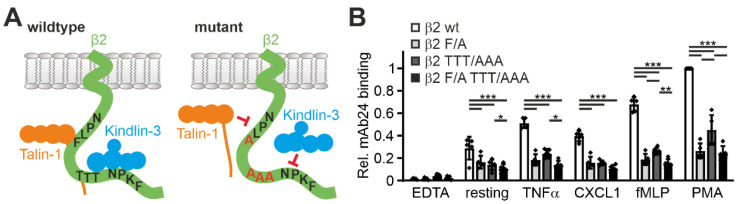
Talin and/or kindlin3 binding-deficient human integrin β2 exhibit impaired integrin activation. (**A**) Scheme of the integrin β2 cytoplasmic domain, which binds talin1 via a membrane proximal NPLF motif and kindlin3 via a membrane-distal NPKF and a TTT motif (**left**). Mutation of these motifs to NPLA and AAA prevent talin1 and kindlin3 binding, respectively (**right**). (**B**) Relative mAb24 binding to neutrophil-like cells derived from Hoxb8 cells expressing human wild-type integrin β2, mutant integrin β2 F/A (F754A), mutant integrin β2 TTT/AAA (T758A, T759A, T760A), or double-mutant integrin F/A TTT/AAA in response to TNFα, CXCL1, fMLP, and PMA or left untreated. N = 5 experiments indicated as individual data points. All values are given as mean ± SD. * *p* < 0.05, ** *p* < 0.01, *** *p* < 0.001.

**Figure 7 cells-11-01532-f007:**
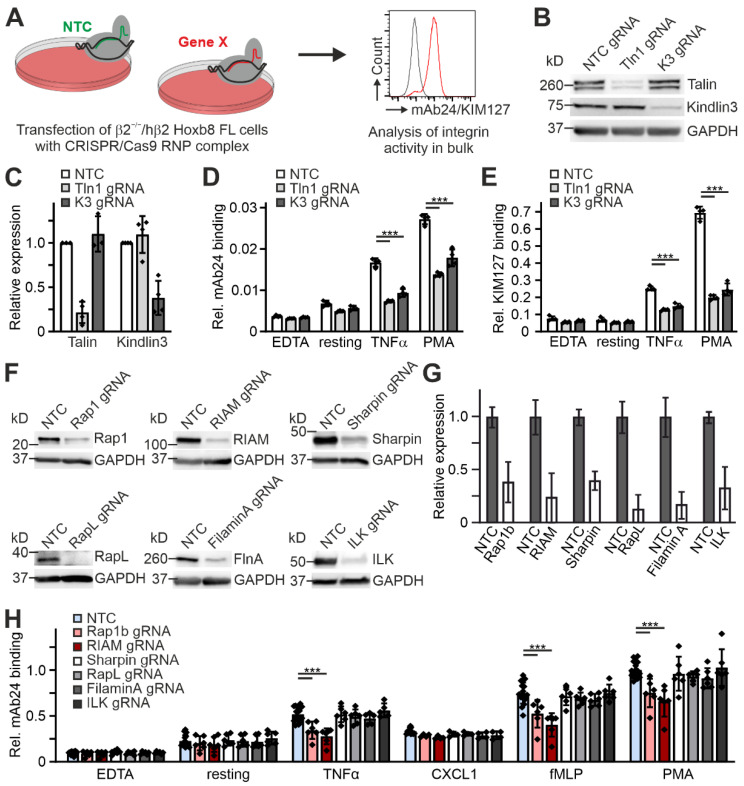
Rapid CRISPR-based screening approach to study putative β2 integrin regulators. (**A**) Scheme of workflow. Integrin β2^−/−^ Hoxb8 cells expressing human integrin β2 were electroporated with a control Cas9/gRNA complex (non-targeting control, NTC) or a Cas9/gRNA complex targeting a gene of interest. Cells were analyzed for integrin activity in bulk after differentiation into neutrophil-like cells. (**B**) Western blot analysis of talin1 and kindlin3 expression in Hoxb8-derived neutrophil-like cells transfected with Cas9 RNP loaded with either a non-targeting (NTC), a talin1-targeting (Tln1 gRNA), or a kindlin3-targeting (K3 gRNA) guide RNA. GAPDH served as loading control. (**C**) Densitometric quantification of talin1 and kindlin3 expression exemplarily shown in (**B**). N = 4 transfections. (**D**,**E**) Integrin β2 activation of the cells described in (**B**) assessed by FACS as relative mAb24 (**D**) and KIM127 (**E**) binding. Cells were either left untreated or treated with EDTA, TNFα, or PMA. N = 4 transfections. (**F**) Western blot analysis of Rap1, RIAM, Sharpin, RapL, Filamin A, and ILK expression in Hoxb8-derived neutrophil-like cells transfected with Cas9 RNP loaded with either a non-targeting RNA (NTC) or a guide RNA targeting Rap1, RIAM, Sharpin, RapL, Filamin A, or ILK. GAPDH served as loading control. (**G**) Densitometric quantification of Rap1, RIAM, Sharpin, RapL, Filamin A, and ILK expression exemplarily shown in (**F**). N = 6 transfections. (**H**) Integrin β2 activation of the cells described in (**F**) assessed by FACS as relative mAb24 binding. Cells were either untreated or treated with EDTA, TNFα, CXCL1, fMLP, or PMA. N = 6 transfections. The data points represent the results of the individual experiments. Values are given as mean ± SD. *** *p* < 0.001.

## Data Availability

The data presented in this study are available on request from the corresponding author.

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
