# Peer review of "Humanized β2 Integrin-Expressing Hoxb8 Cells Serve as Model to Study Integrin Activation"

_cells, 2022, doi:10.3390/cells11091532_

Round 1

Reviewer 1 Report

Hoxb8 cells, which are conditionally immortalized progenitor cells of the hematopoietic system, can be efficiently differentiated into various myeloid and lymphoid cell types in vitro and in vivo. In addition, these cells can be easily generated from the bone marrow of any existing mouse strain. Here, Bromberger et. al. describes the generation of a murine hematopoietic progenitor Hoxb8 cell line expressing human β2 integrin that functionally replace the deleted mouse ortholog with the aim to establish a cell model for rapid testing the role of putative integrin regulators in controlling β2 integrin activity in a physiological context.

The motivation of this research is that available cell models do not express β2 integrins and so do not properly respond to activation triggers. Authors humanized murine Hoxb8 cells from β2 integrin deficient mice by stably expressing the human or murine β2 integrin ortholog, which functionally fully replaced the murine counterpart in PMN-like cells and macrophages.

They show that humanized β2 PMN-like cells can be activated by various agonists such as TNFα, CXCL1, fMLP or PMA that trigger integrin inside-out signaling pathways leading to integrin activation. Further, they exposed humanized β2 PMN-like cells to the same agonists and simultaneously treated them with inhibitors of phosphatidylinositol-3-kinases (PI3K), phospholipase C 331 (PLC) and protein kinase C (PKC), which are central signaling nodes that control β2 integrin activation. These experiments demonstrated that their inhibition significantly impaired β2-integrin activation in humanized PMN-LCs.

Using manipulation via CRISPR/Cas system or by introducing specific β2 integrin mutants via retroviral vectors they show that kindlin3 is required for β2 integrin ectodomain extension and full activation similar to talin1. This is the most important result of the paper and authors propose revision of the proposed model of sequential binding of talin and kindlin i.e. where binding of talin1 to the integrin tail first triggers extension of the integrin ectodomain with intermediate affinity and subsequent binding of kindlin3 leads to full activation. Authors suggest that initial induction of ectodomain extension already requires talin1 and kindlin3 cooperativity.

Authors also showed that key talin regulators Rap1b and RIAM are critical for αLβ2 integrin activation, while other integrin regulatory proteins such as filamin, ILK, RapL and Sharpin are not required.

Authors conclude that the cell system constructed and tested in this article allows rapid testing of mutant β2 integrins or study the role of potential integrin regulators on inducing conformational changes of β2 integrins under physiological conditions.

This is an elegant study tackling the most important proteins regulating integrin activation. Experiments are clearly described and further explained with the schemes in the figures showing corresponding results. The paper is very well written, concise, and the discussion is targeted and clearly emphasizes the most important results. Therefore, I recommend acceptance of this article in the present form.

Author Response

We are very pleased that the reviewer is satisfied with our manuscript and thank him/her for the very positive evaluation.

Reviewer 2 Report

The study by Bromberger and colleagues describes an innovative cell system for probing b2 integrin activation mechanisms using chimeric human integrin expression in murine HoxB8 progenitors from which one can derive primary-like neutrophils. It is impressive that the authors observed very similar expression levels of the chimeric b2 integrins as on wildtype neutrophils. This approach now enables the use of two activation epitope-specific antibodies in neutrophils that can be genetically manipulated (as progenitors). Interestingly, they found that under certain conditions that kindlin3 may play a greater role than previously appreciated in the conversion of bent b2 integrins to their extended conformation reported by the KIM127 mAb. The experiments are generally well done and most of the conclusions of the study are supported by the data. The questions and concerns I have are:

  1. For analyses of outside-in signaling (Figure 2E), because the b2-/- neutrophils are not adherent at all, it weakens the interpretation that the human/mouse chimeric b2 integrin is able to transduce outside-in signals equal in nature to the endogenous murine b2 integrin – the phosphoproteins probed may be induced by cell adhesion generally. To more specifically demonstrate b2 integrin-mediated signaling, the authors could consider a dual substrate composed of a non-integrin ligand (e.g. poly-L-lysine) combined with ICAM-1 so that b2-/- neutrophils would adhere but not signal through b2 integrins.
  2. The experiment shown in Figure 4D probes KIM127 binding to several control/knockout lines of neutrophils and indicates that both talin1 and kindlin3 deficiency reduces b2 integrin extension in response to either TNFa or PMA as soluble stimuli. Based on these data, the authors’ conclude on line 353 that “both talin1 and kindlin3 are required for b2 integrin extension.” In this reviewer’s opinion, this broad conclusion is an over-interpretation of the data, especially since (with respect to the role of kindlin3) it is in contradiction to previous studies, as they acknowledge in the Discussion (paragraph starting at line 493). One alternate interpretation is that it may be that b2 integrin extension with certain soluble stimuli requires kindlin3, but under other conditions it is not. The authors should account for such considerations in interpreting their data. While the authors may question whether extended b2 integrins mediate “slow rolling”, there are multiple lines of evidence that point to such, including induction of KIM127 (see Kuwano et al 2010 Blood).
  3. Relatedly, Figure S3B may also be in contradiction to the conclusion on line 353, as this data suggests that b2 integrin-dependent rolling is at least partially intact in kindlin3-deficient neutrophils. I would highly recommend including these data in the main figures of the paper. Demonstrating that the difference in rolling velocity between wildtype and kindlin3-deficient neutrophils is due to b2 integrin-mediated interactions (e.g., using anti-CD18) would also be an important control, as other factors (cell size, selectin-mediated rolling, inclusion of CXCL1 as stimulus). I suspect that the difference between wildtype and kindlin3 knockout neutrophils in the assay as it was performed may be due to P-selectin-stimulated neutrophil “slow rolling” remaining fully intact in the absence of kindlin3, but CXCL1-stimulated “slower rolling” (eventually leading to stable adhesion) being deficient in the absence of kindlin3.
  4. On line 484, the authors point to the derivation of Hoxb8 cells from existing mouse strains as an approach to “avoiding clonal artefacts”, but may want to revise this statement since there is certainly some variation between each independent Hoxb8 cell line derivation. While these may not be clonal artifacts, they are artifacts nonetheless.

Reviewer 3 Report

In this report, the Authors established a cell model, murine immortalized hematopoietic progenitor Hoxb8 cells, capable of differentiation into neutrophil-like cells, macrophages or dendritic cell. To this end, the Authors generated Hoxb8 cells expressing human β2 integrin instead of the mouse ortholog, by retroviral expression of human β2 integrin in mouse β2 integrin null background.  These cells are readily stimulated by physiological stimuli; the outside-in β2 integrin signaling can be detected by the conformation-specific antibodies against human β2, mAb24 and KIM127.  Furthermore, these cells can be efficiently manipulated via the CRISPR/Cas9 mediated knock-down of specific proteins. The Authors showed that CRISPR/Cas9 mediated kindlin-3 or talin deletion or the expression of integrin with the mutations in the cytoplasmic tail, that prevent kindlin-3 or talin binding hampered both β2 integrin extension and full activation, as measured by antibody binding. Moreover, humanized Hoxb8 cells can be used to rapidly screen for new putative β2 integrin activators. Overall the study is well designed and of interest to the readers. The Hoxb8 system has several advantages over the previously used model systems to study integrin activation (like A5-CHO cells). There cells are not only can be isolated and immortalized from any existing mouse strain and genetically manipulated, also after in vitro differentiation they mimic functionally analogous primary myeloid cells.     

Minor comments:

  1. Figures 2E, 4B, 5B, molecular weight markets are missing.
  2. How the Authors explain the activation of β2 integrin upon PMA stimulation in kindlin-3 deficient cells? Can they comment on the possible mechanism?
  3. In Figure 5H, the Authors show that ILK knock-down has no effect on integrin activation. The recent publication (Margraf, A. et.al. Blood 2020, 136 (19), 2200-2205), the Authors established that ILK was essential for kindlin-3 mediated integrin activation in murine neutrophils. Why the discrepancy?               

Author Response

We thank the reviewer for his/her positive evaluation and his/her comments:

  1. Figures 2E, 4B, 5B, molecular weight markets are missing.

We added the weight markers

  1. How the Authors explain the activation of β2 integrin upon PMA stimulation in kindlin-3 deficient cells? Can they comment on the possible mechanism?

We believe the reviewer refers to Figure 4C, where we measured a slightly higher (but also significant) mAb24 binding in kindlin3 deficient neutrophils compared with talin1 deficient neutrophils. We would not like to overinterpret these data because, first, PMA is a very potent, non-physiological agonist and, second, this effect is not observed after TNFa stimulation. It could be that PMA triggers massive talin1 membrane recruitment and integrin binding that induces a detectable increase in b2 integrin ectodomain extension, which might be in equilibrium with the high affinity conformation.

  1. In Figure 5H, the Authors show that ILK knock-down has no effect on integrin activation. The recent publication (Margraf, A. et.al. Blood 2020, 136 (19), 2200-2205), the Authors established that ILK was essential for kindlin-3 mediated integrin activation in murine neutrophils. Why the discrepancy?               

This is indeed a surprising finding that puzzled us as well. Several reasons might account for the difference. First, as we also mentioned in the text, we only analysed bulk ILK knockout cells with a knockout efficiency of appr. 70% (see Figure 7F,G (former 5F,G)), which might not be enough to see the integrin activation defect. Second, and this is our preferred interpretation, in the study of Margraf et al the authors coated the Kim127 and mAb24 antibodies in their flow chambers and counted arrested cells. While this kind of experiment indeed indicates that the arrested cells express integrins on their surface with a certain integrin conformation (which is detected by the two antibodies), cell arrest still requires strengthening of the integrin-cytoskeleton interaction, which we believe is ILK dependent.

Round 2

Reviewer 2 Report

The authors have addressed all of my questions and critiques.